# Feelings, Behavioral Actions and Depressive Symptoms Related to COVID-19 among Undergraduates in Hail, Saudi Arabia

**DOI:** 10.3390/healthcare9101280

**Published:** 2021-09-28

**Authors:** Badr K. Aldhmadi, Ramaiah Itumalla, Rakesh Kumar, Bilesha Perera

**Affiliations:** Department of Health Management, College of Public Health and Health Informatics, University of Hail, Hail P.O. Box 2440, Saudi Arabia; r.itumalla@uoh.edu.sa (R.I.); ra.kumar@uoh.edu.sa (R.K.); bileshap@gmail.com (B.P.)

**Keywords:** COVID–19, depressive symptomatology, feelings, behavioral acts, undergraduates

## Abstract

The severe acute respiratory syndrome coronavirus 2 (COVID-19) outbreak has had a profound psychological impact on university undergraduates. Feelings, behavioral actions, and depressive symptoms related to COVID-19 in undergraduates in Hail, Saudi Arabia, were assessed using an online survey. Eighteen feelings and six behavioral acts were assessed. The Center for Epidemiological Studies Depression Scale was used to measure depressive symptomatology. Descriptive statistics and logistic regression techniques were used. The mean age of the participants (*n* = 418) was 20.2 years (standard deviation (*SD*) = 1.8 years), and 52.9% (*n* = 221) were males. Elevated levels of depressive symptoms were reported by 47.1% of male and 51.3% of female participants. Social isolation, loss of interest, obsessive monitoring of symptoms, concentration difficulties, recurrent negative thoughts, and worries about health services emerged as vital negative feelings related to COVID-19 that were expressed by the participants. Younger age (odds ratio (OR) = 0.636, 95% confidence interval (95% CI) = 0.428–0.946) and coming from the middle-income category (OR = 0.388, 95% CI = 0.151–0.994) were found to be protective factors against developing depressive symptoms. Frequent cleaning of hands, wearing masks when going out, and adherence to social distancing rules were practiced by 71.5% (95% CI = 67.2–75.8%), 78.7% (95% CI = (74.4–82.5%) and 66.0% (95% CI = (61.3–70.5%) of the participants, respectively. These behavioral acts were not associated with the development of depressive symptoms. Although the practice of COVID-19 precautionary measures by the participants was satisfactory, nearly half of the participants reported depressive symptoms. Innovative educational strategies are needed to curb concentration difficulties and social isolation experienced by undergraduates during outbreaks such as COVID-19.

## 1. Background

The severe acute respiratory syndrome coronavirus 2 (SARS-CoV-2 or COVID-19) pandemic is currently threatening the lives of millions of individuals worldwide [1,2,3]. The World Health Organization (WHO) declared COVID-19 a pandemic on 11 March 2020. By 12 June 2021, 176,041,306 confirmed COVID-19 cases and 3,800,595 deaths had been recorded worldwide [4]. In addition to COVID-19-related deaths, myriad health, psychosocial, and economic consequences of the pandemic are rapidly emerging in many countries [1,5,6]. The unprecedented spread of COVID-19 has forced these countries to implement strict control measures to mitigate the impact of COVID-19 on human lives. City-wide lockdowns, curfews, closure of schools and universities, transportation restrictions, community screening, and the promotion of the adoption of safety measures are some of the control strategies that have helped those countries curb the explosive spread of COVID-19 [3,7].

The COVID-19 outbreak has significant indirect effects on the academic and psychosocial functioning of undergraduates [2,8,9]. An online survey conducted among 44,447 university students in China in early February 2020 found a considerable prevalence of anxiety (7.7%) and depressive symptoms (12.2%) [10]. Another study conducted among 2031 university students in the United States in May 2020 found that 48.14% and 38.48% of the participants had moderate-to-severe levels of depression and anxiety, respectively [11]. It was observed that the fear of contracting COVID-19, the rapid spread of the deadly virus, the shortage of medical and other personal protective equipment, financial losses, restrictions in mobility, and disruptions to educational activities, including uncertainties about future job perspectives, were some of the vital factors threatening the social and psychological well-being of university students during the pandemic. These adversities could have caused many of them to develop depression, fatigue, anxiety, and loss of interest [10,11,12].

The first COVID-19 case in Saudi Arabia was reported on 2 March 2020 [13]. Subsequently, rigorous policy actions were implemented by the government to control the spread of the epidemic in university communities. By 12 March, all social and governmental gatherings and events were suspended or postponed. Schools and universities were closed, remote modes of learning and teaching were introduced, and travel restrictions were implemented [9,14,15]. Online teaching in the universities continued till the end of the year 2020.These sudden changes in the mode of education may have disrupted the living patterns of the students, resulting in uneasiness, frustration, and escalating internal distress. Factors related to psychological well-being in Saudi university students during a serious disease outbreak, such as that of COVID-19, are also shaped by cultural and socioeconomic factors. Countries in the Gulf region exhibit somewhat unique sociocultural living patterns compared with the rest of the world, such as the salience of religion in daily life and segregation of men’s and women’s educations. Thus, culture-sensitive and gender-sensitive guidelines for effectively facing adversities related to COVID-19 and regulating adverse feelings and emotions that arise during the pandemic are needed. Research in this area is a prerequisite to developing effective health and educational guidelines. Nevertheless, several measures have been implemented by the Saudi government since July 2020 to ease the performance of day-to-day activities by the general public. Curfews and lockdowns were lifted, but large gatherings of individuals were prohibited. The wearing of masks was made compulsory, and universities across the Kingdom have been reopened with online teaching, with some exams being conducted in person [16].

This study aimed to determine the prevalence of depressive symptoms and to examine feelings and behavioral actions related to COVID-19 among undergraduates in Hail, Saudi Arabia. Additionally, we examined the relationship between depressive symptoms and the practice of precautionary behavioral acts in undergraduates.

## 2. Methods

### 2.1. Study Setting and Design

The target population of the study was undergraduates at Hail University in Saudi Arabia. A cross-sectional online survey was conducted. Data collection was performed between 15 August and 15 September 2020. An anonymous questionnaire was used for the survey to ensure the confidentiality and reliability of the data.

#### Study Instruments and Variables

The study instrument was an online survey questionnaire. The survey questionnaire assessed the following demographic characteristics: age, gender, year of study, and monthly family income. The questionnaire also included 18 items that assessed participants’ feelings related to the COVID-19 outbreak, and responses were recorded on a 5-point scale (1 = yes, frequently to 5 = no, never). Items such as “The COVID-19 outbreak has made me anxious”, “The COVID-19 outbreak has made me wear masks”, “The COVID-19 outbreak has made me suffer from social isolation”, and “The COVID-19 outbreak has made me lose interest” were included. These items were selected based on an extensive literature review [2,7,8,9,10,17]. A pilot test of the comprehensibility and understandability of the 18 items was performed with 5 undergraduates at the same university. Based on their comments, the wording of the questions was improved. The Cronbach’s alpha of the 18 items was 0.812. Depressive symptoms were assessed using the Center for Epidemiological Studies Depression Scale (CES-D), which was developed to screen for depression in community settings [18]. The CES-D is a reliable and valid scale that has been widely used in surveys conducted among adolescents and young adults [19,20]. Total scores can range from 0 to 60, and the cutoff score is 16. Those with scores ≥ 16 are considered to have elevated levels of depressive symptoms. The Cronbach’s alpha value of the CES-D scale was 0.898. The practice of six precautionary behaviors related to COVID-19 was assessed using items such as “I wash my hands or use sanitizer frequently” and “I have limited my travelling”, which were rated on a 5-point scale (1 = yes, frequently to 5 = no, not at all).

The questionnaire was developed in English and subsequently translated into Arabic. Both versions of the survey were used to ensure that accurate responses were elicited from the participants. The final draft of the online survey form was piloted using 10 undergraduates at the same university, and based on the results, the final survey form was obtained.

### 2.2. Sampling Procedure

Probability sampling techniques could not be used to select subjects because classroom teaching had been suspended in Saudi Arabia at the time of data collection. Therefore, an electronic platform was used, and the questionnaire was sent to students electronically. Snowball sampling was used to recruit as many respondents as possible. Through our professional and personal networks, we contacted class representatives and requested that they forward the survey link to the undergraduate students at Hail University through WhatsApp messenger. Attempts were made to obtain an equal number of male and female undergraduates for the survey, as approximately 50% of the undergraduates in the university were male undergraduates. Since there were no previous studies done on the depressive symptomatology of university undergraduates during a serious pandemic such as COVID-19, the minimum sample size needed was calculated assuming that 50% of the respondents had elevated depressive symptoms and using a 95% confidence level with a margin of error of 5%. Thus, the minimum sample size required was 384.

### 2.3. Ethical Approval

This study was approved by the Ethics Review Committee of the University of Hail, Saudi Arabia (Ethics Review Number: H-2020-093).

### 2.4. Data Analysis

Statistical Package for Social Sciences (SPSS) version 25 (IBM, Armonk, NY, USA) was used to conduct statistical analyses. The level of statistical significance was set as *p* < 0.05. Frequencies and percentages were computed to examine categorical variables and means and standard deviations were computed to examine continuous variables. Binary logistic regression analysis was performed to identify predictors of increased levels of depressive symptoms among the participants. Three vital personal factors (gender, age and family income) and three major precautionary behavioral acts promoted by the WHO in controlling the spread of COVID-19 (regular hand sanitization, wearing masks, and adherence to social distancing rules) were considered predictors.

## 3. Results

A total of 418 undergraduate students participated in this study. Their demographic characteristics are presented in Table 1. Their mean age was 20.2 years (standard deviation (SD) = 1.8 years). The sample consisted of 221 (52.9%) men and 197 (47.1%) women, which is approximately proportional to the sex ratio of the undergraduates at Hail University. Furthermore, 226 (54.1%) participants were first- or second-year students, and 192 (45.9%) participants were third- or fourth-year students. Thus, an approximately equal proportion of junior and senior undergraduates was sampled. The distribution of monthly family income in Saudi Riyal (SAR) was as follows: SAR < 20,000 = 314 (75.1%), SAR 20,000–50,000 = 75 (17.9%), and SAR > 50,001 = 29 (6.9%).

The participants had a mean score of 17.63 (*SD* = 11.57) on the CES-D scale. Women (18.56) had a slightly higher mean score than men (16.80), but the difference was not significant (*p* = 0.21). The overall prevalence of elevated depressive symptoms was 49.0%. Among male undergraduates, 47.1% reported elevated depressive symptoms, and the corresponding figure for female undergraduates was 51.3%. However, the difference was not significant (*p* = 0.39).

Table 2 presents the distribution of participant responses to items that assessed feelings related to the COVID-19 outbreak. Loss of interest, sense of isolation, obsessive monitoring of symptoms, concentration difficulties, and recurrent negative thoughts about COVID-19 were the most common negative feelings reported by the participants.

Binary logistic regression analysis was conducted to identify the predictors of increased levels of depressive symptoms among the participants. Basic assumptions of the binary logistic regression technique, including that the dependent variable was binary, the observations were independent of each other, no multicollinearity was present among the independent variables, and the independent variables and log odds were linear, were met by the data set.

The logistic regression results are presented in Table 3. In the final model, younger age (odds ratio (OR) = 0.636, 95% confidence interval (CI): 0.43–0.94) and middle family income (OR = 0.38, 95% CI: 0.15, 0.99) emerged as protective factors against depressive symptomatology. Frequent hand sanitization, wearing masks when going out, and adherence to social distancing rules were not associated with depressive symptomatology.

Figure 1 illustrates how the participants adhered to precautionary behavioral acts. The prevalence of high adherence rates was high for cleaning hands (71.5%), wearing masks (78.7%), social distancing (66.0%), and covering the nose when sneezing (84.0%). Somewhat low adherence rates were found for limiting travel (62.9%), indicating that social isolation is a stressful behavior for this population group.

## 4. Discussion

The COVID-19 outbreak has made a significant number of undergraduate students around the world vulnerable to developing a variety of psychopathologies [10,11,21]. Public education about preventive measures, travel restrictions, detection of cases, and treatments for the affected were the main strategies adopted by the Saudi government to control the pandemic [12,13,16]. There is a need to proactively address the psychosocial and academic consequences of COVID-19 in undergraduate students in Saudi Arabia. However, a lack of research findings on factors that are associated with the pathologies caused by COVID-19 among undergraduate students is evident [22,23]. Up-to-date scientific evidence on the impact of COVID-19 on undergraduates and on the higher education system in Saudi Arabia would tremendously help health and educational authorities formulate and implement effective health and educational policies and strategies to mitigate adverse psychological outcomes from the ongoing COVID-19 pandemic in undergraduates.

Approximately 49% of the participants in this study reported elevated depressive symptoms. A study conducted in 2012–2013 in a large university in Saudi Arabia reported the prevalence of depressive symptoms to be 46% [24]. Thus, a slightly higher prevalence was observed in our sample. A study conducted among 1165 university students in Jordan in March 2020 found that 61.4% had moderate-to-severe levels of depression [25]. A study conducted among 476 undergraduate students in Bangladesh in May 2020 found that 82.4% had mild-to-severe depressive symptoms [26]. Therefore, compared with the rates during the COVID pandemic, the prevalence rate of depressive symptoms in our sample was low. However, the timing of the study may have influenced the prevalence because the psychological impact was expected to be much more intense at the beginning of the COVID-19 pandemic. In addition, Saudi Arabia has a well-established healthcare system with ample emergency medical treatment facilities; therefore, undergraduates in Saudi Arabia probably have fewer worries about safety measures available regarding COVID-19 in the country. Moreover, putting strict health protocols into action, such as hand washing, using face masks, advising people to “stay at home”, and closing mosques, probably would have increased their levels of confidence of safety and lowered their stress levels over time. No gender difference in depressive symptoms was observed in our study (male (47.1%) versus female (51.3%)). Our results are in line with some studies [27], but several studies indicated that female undergraduates were more prone to psychological health problems related to COVID-19 than male undergraduates [28,29,30]. Possible reasons for females being more vulnerable to psychological stressors include their higher perceived risk of COVID-19 compared with male students and additional responsibilities associated with caregiving work that they have to take on during the pandemic. However, female undergraduate students in Saudi Arabia are largely limited in terms of mobility and social interactions. Thus, their self-confidence about natural protection against the pandemic may also have contributed to lower their distress levels. Further longitudinal and qualitative research, however, is needed to confirm our assertions.

Binary logistic regression suggests that junior students tended to have fewer depressive symptoms (OR = 0.636 (95% CI = 0.428–0.946)). This finding is consistent with similar studies conducted in other university undergraduate populations [26,27]. In general, senior undergraduates are highly concerned about timely completion of their degrees and future job opportunities. The current pandemic situation has jeopardized senior undergraduates’ ambitions and future perspectives. This may have caused significant deterioration in their psychological health status.

Our observation that 69% of the participants coming from higher income brackets, 41.3% of the participants coming from middle income brackets, and 49% of the participants coming from lower income brackets had elevated depressive symptoms indicates that undergraduates in high income brackets are more likely to develop depressive symptoms than others. Students from affluent families appear to have perceived the current situation more negatively. This pandemic has placed extreme financial pressure on families. A disease outbreak could lead to the loss of sources of income in both rich and poor families and cause students to feel distressed. Nevertheless, outbreaks such as COVID-19 hit large-scale businesses and luxury lifestyle commodities very hard, which would probably affect the minds and lives of people in affluent families more than those of others. Rich people may have been suffering from a sharp decline in income due to the pandemic, as all large-scale businesses and financial transactions were halted during the initial period of the pandemic. This could be the reason for our observation that undergraduates from affluent families were more prone to worry about COVID-19. Furthermore, in the current study, approximately one-third of the participants were worried about the economic hardships that they and their country will have to face due to the pandemic. They may have thought that the recovery of the Saudi economy from the recent financial crises related to oil production and sales would again be affected by the global COVID-19 pandemic. These emotionally exhausting thoughts may have stemmed from a sense of uncertainty about future life, income security, and employment. Previous studies [14,25,31] suggest that income stability is a protective factor that can enhance the psychological well-being of undergraduate students during the COVID-19 pandemic. A study conducted in China found that undergraduate students from families with stable incomes were less likely to develop anxiety symptoms [32]. In Saudi Arabia, compared with the rich, the poor may perceive that their income is stable. However, these discrepant findings underscore the need for further qualitative investigations into the economic impact of the pandemic on the psychological well-being of undergraduate students.

The feelings expressed by undergraduates in the midst of the COVID-19 outbreak are vital for us to identify factors associated with the health and well-being of undergraduates in future waves of the COVID-19 virus. Stress, frustration, fear, worry, anxiety, uncertainty, hope, and gratitude are some of the important feelings experienced by undergraduates to which educational and health authorities need to attend to mitigate the negative consequences of serious infectious diseases. In this study, approximately 83% of the participants reported a sense of isolation, and 77% reported loss of interest at least to a moderate level during the COVID-19 outbreak, as has been seen in other studies [11,26,28]. Wang et al. [10] reported that changes in social relationships, social isolation, and social or physical distancing were among the major lifestyle concerns reported by undergraduate students during the COVID-19 outbreak.

Approximately 74% of the participants were worried about available health facilities. Studies have identified various barriers in undergraduates’ accessibility to mental health services during the pandemic [21,32,33]. Economic hardships and traveling restrictions were the main barriers identified with this regard. Past findings suggest that factors such as reluctance to visit doctors for non-COVID-19-related issues and the social stigma attached to undergoing COVID-19 testing are some of the stressors that undergraduate students have to face during the COVID-19 pandemic [10,26,34]. Thus, it is of paramount importance to remedy these lacunas in future infectious outbreaks. Special attention needs to be given to female undergraduates and undergraduates with a history of psychiatric illnesses, as they are highly vulnerable to these psychological adversities.

It is quite normal that individuals try to determine whether they are exhibiting the signs and symptoms of COVID-19 and search for relevant information by themselves. Approximately 85% of the participants in this study reported that they obsessively monitored the emergence of COVID-19 symptoms to at least a moderate level. The proliferation of misinformation through social media platforms can lead to feelings of uncertainty and indecisiveness among undergraduate students [35,36]. In this regard, digital learning packages that enhance psychological well-being [37] and internet-based psychological therapies will be effective in reducing the stress and psychopathological symptoms caused by the COVID-19 outbreak. Timely dissemination of valid and authenticated information will reduce the number of negative thoughts that emerge in the minds of undergraduate students. Of note, 82.3% of the participants reported experiencing recurrent negative thoughts about COVID-19 to at least a moderate level during the initial period of the pandemic. Recurrent negative thoughts could lead many undergraduates to develop severe psychopathologies, including suicidal ideation. An inability to concentrate was another major negative feeling reported. Approximately 78% of the participants had experienced this feeling at least moderately. Concentration difficulties significantly affect academic performance. Difficulties in adapting to distance learning, coping with an increased workload, and adopting new methods of completing online assignments that require advanced writing skills may have caused a decrease in concentration ability in this vulnerable group [14,38].

The combined effects of these negative feelings foster the development of psychopathologies, including depression, obsessive and impulsive behaviors, and suicidal ideation. Saudi Arabians tend to be a socially and physically active population group. Contingency plans that aim to promote psychological health among undergraduate students during unprecedented events such as the COVID-19 pandemic should be developed using a multisectoral approach and considering the norms, values, and beliefs of Saudi Arabian culture.

A greater proportion of the participants reported that they had been practicing the major precautionary behaviors; 71.5% cleaned their hands frequently, 78.7% wore a mask when going out, and 66% adhered to social distancing rules. It is interesting to note that only 28.7% of the participants tended to read or watch information related to COVID-19. Awareness of facts related to COVID-19 would help this educated segment of the population take necessary precautions to prevent COVID-19. Educational authorities should pay attention to improve the reading habits of this educated population group. Sociocultural norms and practices may make it difficult for Saudi undergraduates, specifically male undergraduates, to isolate themselves from other people and adhere to the rule “stay at home”. However, the logistic regression results indicate that there was no association between depressive symptoms and adherence to precautionary behaviors in the target population.

This study has several limitations. Data were collected using an online questionnaire due to the closure of the universities at the time of the survey. Snowball sampling was used. Furthermore, the survey method did not allow the participants to seek clarifications about the survey or items in the questionnaire. Due to the nature and time constraints associated with online surveys, it was not possible to include items on other important control variables that could influence the relationships between the independent and dependent variables in the logistic regression analysis. Nonetheless, the large sample size and age and gender distributions of the sample indicated that the sample was representative of the population of undergraduate university students in Hail, Saudi Arabia, and therefore that the results obtained were reliable.

## 5. Conclusions

The COVID-19 outbreak has caused in an increase in depressive symptoms in undergraduate students in Hail, Saudi Arabia. The risk of developing these symptoms was similar in the male and female participants. Social isolation, loss of interest, and recurrent negative thoughts related to COVID-19 seem to adversely affect the psychological wellbeing of university undergraduates in Hail, Saudi Arabia. Senior undergraduates and those who came from affluent families seem to have had a higher risk of developing depressive symptoms. The precautionary behavioral acts practiced by undergraduates were satisfactory. Nevertheless, undergraduate students should be educated and equipped to access and acquire accurate information from trustworthy sources, and mental health and counseling services should be readily available for them during emergencies via digital mental health services. Both longitudinal and qualitative studies that consider a wide range of explanatory variables that can explain depressive symptomatology among university students during epidemics such as COVID-19 are warranted.

## Figures and Tables

**Figure 1 healthcare-09-01280-f001:**
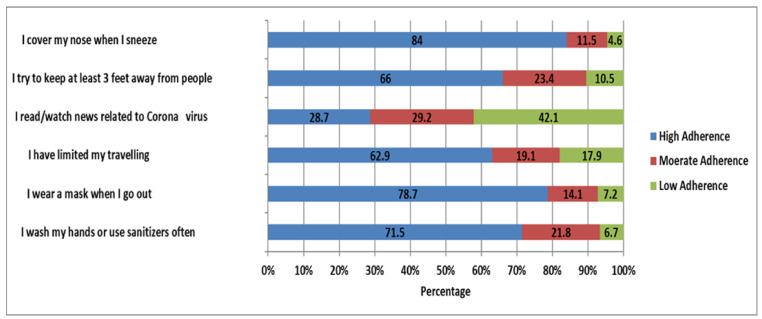
Adoption of Precautionary Behaviors to Prevent Coronavirus Disease Transmission (*n* = 418).

**Table 1 healthcare-09-01280-t001:** Demographic characteristics of the participants (*n* = 418).

Characteristic		Frequency (%)
Age (years)	≤19	188 (45.0)
	20–21	123 (24.4)
	≥22	107 (25.6)
Gender	Male	221 (52.9)
	Female	197 (47.1)
Year of study	First and second year	226 (54.1)
	Third and fourth year	192 (45.9)
Monthly family income (SAR)	≤20,000	314 (75.2)
	20,001–50,000	75 (17.9)
	≥50,001	29 (6.9)

SAR: Saudi Riyal.

**Table 2 healthcare-09-01280-t002:** Feelings related to the coronavirus disease outbreak in the participants (*n* = 418).

Coronavirus Has Made Me	Mean	SD	Yes, Very Frequently	Very Often	Moderately	Rarely	No, Not At All
Anxious	2.90	1.370	94 (22.5%)	64 (15.3%)	115 (27.5%)	78 (18.7%)	67 (16.0%)
Have difficulty studying	3.07	1.383	76 (18.2%)	69 (16.5%)	113 (27.0%)	71 (17.0%)	89 (21.3%)
Avoid interacting with outsiders	3.57	1.281	41 (9.8%)	39 (8.9%)	114 (27.3%)	95 (22.7%)	131 (31.3%)
Feel a loss of interest	2.26	1.298	177 (41.1%)	71 (17.0%)	102 (24.4%)	40 (9.6%)	33 (7.9%)
Feel a sense of isolation	2.39	1.429	171 (41.4%)	65 (15.6%)	89 (19.9%)	46 (11.0%)	53 (12.7%)
Wear masks	4.43	0.994	13 (3.1%)	7 (1.7%)	56 (13.4%)	55 (13.2%)	287 (68.7%)
Wash hands several times a day	4.17	1.142	15 (3.6%)	28 (6.7%)	69 (16.5%)	66 (15.8%)	240 (57.4%)
Have fewer food options	2.77	1.379	111 (26.6%)	63 (15.1%)	118 (28.2%)	65 (15.6%)	61 (14.6%)
Worry about my financial status	2.83	1.446	114 (23.7%)	60 (14.4%)	104 (24.9%)	64 (15.3%)	76 (18.2%)
Worry about my country’s economy	3.23	1.461	83 (19.9%)	44 (10.5%)	99 (23.7%)	77 (18.4%)	115 (27.5%)
Scared about my health	3.96	1.243	28 (6.7%)	26 (6.2%)	85 (20.3%)	75 (17.9%)	204 (48.8%)
Worry about my loved ones	4.42	1.018	15 (3.6%)	9 (2.2%)	48 (11.5%)	58 (13.9%)	288 (68.9%)
Scared about my future	3.49	1.408	60 (14.4%)	36 (8.6%)	107 (25.6%)	70 (16.7%)	145 (34.7%)
Worry about available health services	2.59	1.424	140 (33.5%)	61 (14.6%)	109 (26.1%)	46 (11.0%)	62 (14.8%)
Worry about the future of our people	3.25	1.465	80 (19.1%)	47 (11.2%)	100 (23.9%)	70 (16.7%)	121 (28.9%)
Obsessively monitor symptoms	2.18	1.263	173 (41.4%)	92 (22.0%)	92 (22.0%)	27 (6.5%)	34 (8.1%)
Unable to concentrate	2.45	1.387	147 (35.2%)	81 (19.4%)	98 (23.4%)	37 (8.9%)	55 (13.2%)
Have recurrent negative thoughts	2.17	1.328	189 (45.2%)	80 (19.1%)	74 (17.7%)	38 (9.1%)	37 (8.9%)

SD: standard deviation.

**Table 3 healthcare-09-01280-t003:** Logistic regression analysis results for the predictors of depressive symptomatology.

Predictors	Final Model
	OR (95% CI)
Gender	
Female	1.0
Male	0.945 (0.626–1.426) ^ns^
Age	
≥21 years	1.0
≤20 years	0.636 (0.428–0.946) ^*^
Family income (SR)	
High (>50,000)	1.0
Middle (20,000–50,000) Low (<20,000)	0.388 (0.151–0.994) _*_ 0.506 (0.217–1.182) ^ns^

Frequent hand sanitization	1.078 (0.855–1.358) ^ns^
Wearing masks when going out	1.208 (0.957–1.525) ^ns^
Adherence to social distancing rules	1.006 (0.826–1.225) ^ns^
χ^2^ 18.26 H-L goodness-of-fit 0.818 −2 log likelihood 561.1	

* *p* < 0.05. ns: not significant. OR: Odds ratio. CI: confidence interval. SAR: Saudi Riyal.

## Data Availability

The data sets generated and/or analyzed in the current study are available from the corresponding author and will be provided on reasonable request.

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
