# Peer review of "Feelings, Behavioral Actions and Depressive Symptoms Related to COVID-19 among Undergraduates in Hail, Saudi Arabia"

_healthcare, 2021, doi:10.3390/healthcare9101280_

Round 1

Reviewer 1 Report

19 among undergraduates in Hail, Saudi Arabia” explores issues related to the psychological functioning of the undergraduates from Saudi Arabia that are important in the era of the ongoing coronavirus pandemic.

I highly appreciate the preparation of the article, both in the theoretical and methodological and empirical part. I have some minor comments, which are listed below:

There is no information about the design study - it was probably a cross-sectional web-based study. Please complete it.

I recommend a more detailed description of the measurement tools, including the Cronbach's alpha reliability coefficient for CES-D in the current sample.

Considering the method and method of data collection, if and how the authors controlled that the survey was not filled in twice by the same person.

The authors report that the Ethics Review Committee approved the study, so it can be assumed that it met the criteria for voluntary and informed participation, and participants had the right not to answer certain questions. In that case, were they able to omit the answers in the electronic questionnaire, and if so, were there any missing data, what was their percentage and how did the authors deal with it?

Author Response

  1. 19 among undergraduates in Hail, Saudi Arabia” explores issues related to the psychological functioning of the undergraduates from Saudi Arabia that are important in the era of the ongoing coronavirus pandemic.

Thank You

  1. I highly appreciate the preparation of the article, both in the theoretical and methodological and empirical part.

Thank You

  1. There is no information about the design study - it was probably a cross-sectional web-based study. Please complete it.

Will Do It

  1. I recommend a more detailed description of the measurement tools, including the Cronbach's alpha reliability coefficient for CES-D in the current sample

We Will Add Detailed Description Of The Measurement Tools In The Revised Manuscript

  1. Considering the method and method of data collection, if and how the authors controlled that the survey was not filled in twice by the same person.

We Have Informed The Respondents To Fill The Questionnaire Only Once And Respondents Were Not Allowed To Respond The Same Questionnaire Using The Same Email Id.

  1. The authors report that the Ethics Review Committee approved the study, so it can be assumed that it met the criteria for voluntary and informed participation, and participants had the right not to answer certain questions. In that case, were they able to omit the answers in the electronic questionnaire, and if so, were there any missing data, what was their percentage and how did the authors deal with it?

They Were Allowed To Omit Any Questions. Only Five Respondents Out Of 418 Answered With Less Than 90% Of The Questions, But They All Had Answered More Than 80% Of The Questions. So Missing Data Was Not A Problem In This Study.  

Reviewer 2 Report

  1. The introduction establishes an empirical framework appropriate to the problem.
  2. The contextual frame is made explicit in an adequate way, for example cultural aspects of the region are related to the pandemic.
  3. Please clarify the dates or chronology of health policies by the health system (closing, opening)
  4. Specifically in the survey collection period, please specify health indicators related to COVID. Compare periods and comment based on your results.
  5. Why is only one university sample collected? Comment
  6. please refer to possible role of social desirability in the results

Author Response

  1. The introduction establishes an empirical framework appropriate to the problem.           

Thank You

  1. The contextual frame is made explicit in an adequate way, for example cultural aspects of the region are related to the pandemic

Thank You

  1. Please clarify the dates or chronology of health policies by the health system (closing, opening)

Will Briefly Include These Information In The Text

  1. Specifically in the survey collection period, please specify health indicators related to COVID. Compare periods and comment based on your results

Will Insert These Information In The Revised Manuscript

  1. Why is only one university sample collected? Comment

The Research Was Conducted In Hail, Sa Because The Scope Of The Research Was Confined To Hail Region. We Have Collected Data From The Largest University In Hail. Undergraduate Behaviour Related To Covid 19 In Universities In Other Areas May Be Different From This 

  1. please refer to possible role of social desirability in the results

In Fact No Issue Of Social Desirability Arose As The This Topic Was Not A Sensitive Topic For The Respondents To Have Social Bias When Answering